# Predicting Cutaneous Squamous Cell Carcinoma Progression Risk from Whole Slide Images with Federated Learning

**Jakub Zacharczuk**[*2,3]                    JZ418488@STUDENTS.MIMUW.EDU.PL

**Juan Ignacio Pisula**[*1,3]                    JUAN.PISULA@UK-KOELN.DE

**Doris Helbig**[5]                    DORIS.HELBIG@UK-KOELN.DE

**Lucas Sancéré**[1,3]                    LUCAS.SANCERE@UK-KOELN.DE

**Oana-Diana Persa**[6]                    OANA-DIANA.PERSA@TUM.DE

**Corinna Bürger**[1,7,8]                    CORINNA_BUERGER@YAHOO.DE

**Anne Fröhlich**[9]                    ANNE.FROEHLICH@UKBONN.DE

**Carina Lorenz**[1,7,8]                    CLORENZ6@UNI-KOELN.DE

**Sandra Bingmann**[10]                    SBINGMANN@SKINMED.CH

**Dennis Niebel**[11]                    DENNIS.NIEBEL@KLINIK.UNI-REGENSBURG.DE

**Konstantin Drexler**[11]            KONSTANTIN.DREXLER@KLINIK.UNI-REGENSBURG.DE

**Jennifer Landsberg**[9]                    JENNY.LANDSBERG@UKBONN.DE

**Roman Thomas**[7,12,13]                    ROMAN.THOMAS@UNI-KOELN.DE

**Johannes Brägelmann**[†1,7,8]            JOHANNES.BRAEGELMANN@UNI-KOELN.DE

**Katarzyna Bozek**[†1,3,4]                    K.BOZEK@UNI-KOELN.DE

[1] *Center for Molecular Medicine Cologne (CMMC), Faculty of Medicine and University Hospital Cologne, University of Cologne, Germany*

[2] *Faculty of Mathematics, Informatics and Mechanics, University of Warsaw, Poland*

[3] *Institute for Biomedical Informatics, Faculty of Medicine and University Hospital Cologne, University of Cologne, Germany*

[4] *Cologne Excellence Cluster on Cellular Stress Responses in Aging Associated Diseases (CECAD), University of Cologne, Germany*

[5] *Department for Dermatology, University Hospital Cologne, Cologne, Germany*

[6] *Department of Dermatology, Technical University Munich, Munich, Germany*

[7] *University of Cologne, Faculty of Medicine and University Hospital Cologne, Department of Translational Genomics, Cologne, Germany*

[8] *University of Cologne, Faculty of Medicine and University Hospital Cologne, Mildred Scheel School of Oncology, Cologne, Germany*

[9] *Department of Dermatology and Allergology, University Hospital Bonn, Bonn, Germany*

[10] *Department for Dermatology, University Hospital Cologne, Cologne, Germany*

[11] *Department of Dermatology, University Medical Center Regensburg, Regensburg, Germany*

[12] *Institute of Pathology, Medical Faculty, University Hospital Cologne, University of Cologne, Cologne, Germany*

[13] *DKFZ, German Cancer Research Centre, German Cancer Consortium, Heidelberg, Germany*

**Editors:** Accepted for publication at MIDL 2025

---

[*] Contributed equally

[†] Contributed equally

## Abstract

Cutaneous squamous cell carcinoma (cSCC) is the second most common cancer globally. While surgical excision is typically successful, a significant proportion of patients experience disease progression leading to poor prognosis. Based on the fact that histopathological tumor features have been associated with increased risk of cSCC progression, we propose to predict this condition solely from Whole Slide Image (WSIs) scans of excised tumors. A major challenge in developing such predictive models is the fact that numerous clinical centers maintain patient cohorts that are often too small individually for robust deep learning (DL) applications. Here we use four small to medium-sized datasets from different clinical centers across Germany and demonstrate the feasibility of training federated DL models to predict cSCC progression. We compare various Federated Learning (FL) approaches, leveraging distributed datasets and developing center-specific models.

**Keywords:** Federated Learning, Cutaneous Squamous Cell Carcinoma

## 1. Introduction

According to various estimates, around one million people in the United States are diagnosed with cSCC each year (Jiang et al., 2024; Rogers et al., 2015). Accurate prediction of cSCC progression risk is crucial for determining which patients will benefit from enhanced secondary prevention e.g. by more frequent follow-up care or additional treatments. While existing cSCC staging systems like the American Joint Committee on Cancer (AJCC), the Brigham Women's Hospital (BWH), or the National Comprehensive Cancer Network (NCCN) staging systems provide guidelines on risk stratification and clinical management of cSCC patients (Ruiz et al., 2019b; Schmults et al., 2021; Ruiz et al., 2019a), they fall short of reliably identifying patients at high risk of disease progression. In this work we predict cSCC progression risk directly from Whole Slide Images (WSIs) of excised tumor scans. We train our model with Federated Learning (FL) on a diverse dataset from four clinical cohorts, and adapt personalization strategies to boost local model performance.

## 2. Methodology

**Dataset**  Our WSI dataset was collected from 277 patients with primary cSCC diagnosed and treated by excision at four hospitals in Germany. Progression status was determined via clinico-pathological parameters based on medical records, pathology reports, and active follow-up over many years. Appendix A provides a detailed description of the dataset.

**Method**  We employ a TransMIL (Shao et al., 2021) WSI classifier, using an ImageNet pretrained EfficientNet-v2-L model as feature extractor (Tan and Le, 2021). We experimented with three approaches. In the first place, we trained 4 local models. Next we trained our classifier with the well-established Federated Averaging (FedAvg) strategy (McMahan et al., 2023). Lastly, we employed a simple Personalized FL (PFL) strategy that consist of fine-tuning the global model produced by FedAvg by training on the local cohorts. Additional preliminary results include the use of a SuPerFed (Hahn et al., 2022) strategy, as well as different slide preprocessing steps (Appendix B).

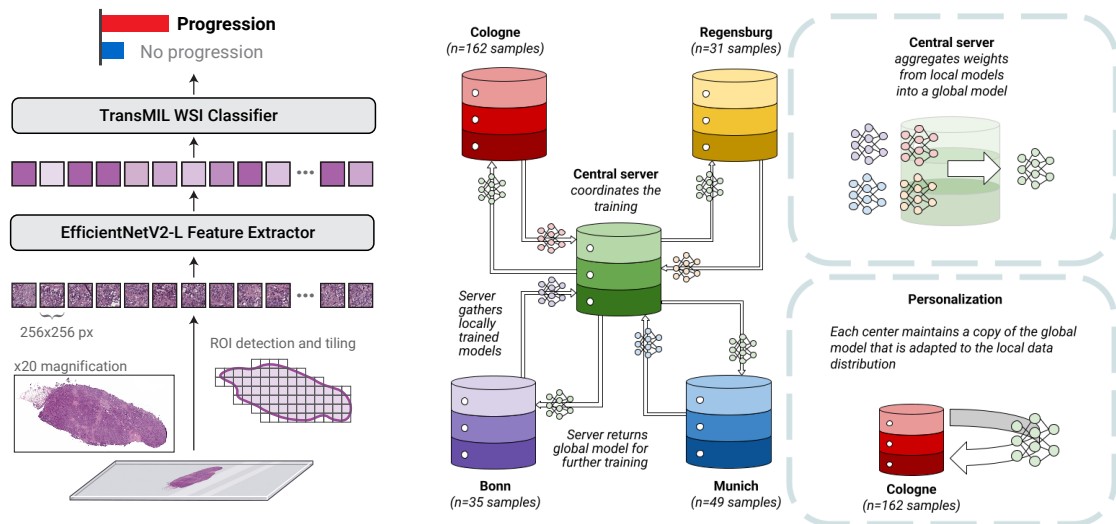

Figure 1: (a) We train a TransMIL model to predict disease progression in patients with cSCC directly from excised tumor biopsies. (b) Our model is trained on a diverse dataset from 4 clinical centers using FL and client personalization.

## 3. Results

Table 1 shows 5-fold cross-validation of the best-performing models in our preliminary experiments. Across all metrics, FedAvg resulted in a performance drop, while the best results were achieved with PFL. Preliminary results and an expanded Table 1 are shown in Appendix B.

| Method | Accuracy | Precision | Recall | F1-score |
|---|---|---|---|---|
| Local models | 0.70 ($\sigma$=0.12) | 0.60 ($\sigma$=0.23) | 0.63 ($\sigma$=0.27) | 0.56 ($\sigma$=0.17) |
| **(FL)** FedAvg | 0.60 ($\sigma$=0.25) | 0.46 ($\sigma$=0.35) | 0.47 ($\sigma$=0.33) | 0.40 ($\sigma$=0.28) |
| **(PFL)** FedAvg+FT | **0.82** ($\sigma$=0.11) | **0.74** ($\sigma$=0.11) | **0.71** ($\sigma$=0.28) | **0.70** ($\sigma$=0.26) |

Table 1: Averaged prediction results metrics over all centers using 5-fold cross-validation.

## 4. Discussion

This work shows that we can predict cSCC patients with high risk of developing disease progression directly from histopathology slides by employing a personalized federated training. We believe that the initial performance drop of FedAvg is caused by domain shift between the cohorts due to e.g. differences in tissue processing. However, our work demonstrates that privacy-preserving applications can leverage diverse and limited-sized datasets by employing center personalization strategies, even outperforming local models.

## Acknowledgments

We furthermore thank the Regional Computing Center of the University of Cologne (RRZK) for providing computing time on the DFG-funded (Funding number: INST 216/512/1FUGG) High Performance Computing (HPC) system CHEOPS as well as support.

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

# Appendix A. Dataset description

Our WSI dataset was collected from patients with primary cSCC diagnosed and treated by excision at four hospitals in Germany. Progression status was determined via clinico-pathological parameters based on medical records, pathology reports, and active follow-up over many years. WSIs were acquired from Hematoxylin-Eosin (H&E) slides using a NanoZoomer Slide Scanner (Hamamatsu) at 40x resolution.

In total, the final dataset used for training the DL models consisted of 277 patients: 162 patients from the Cologne cohort, 35 patients from the Bonn cohort, 49 patients from the Munich cohort, and 31 patients from the Regensburg cohort. The data distribution is shown in Figure 2.

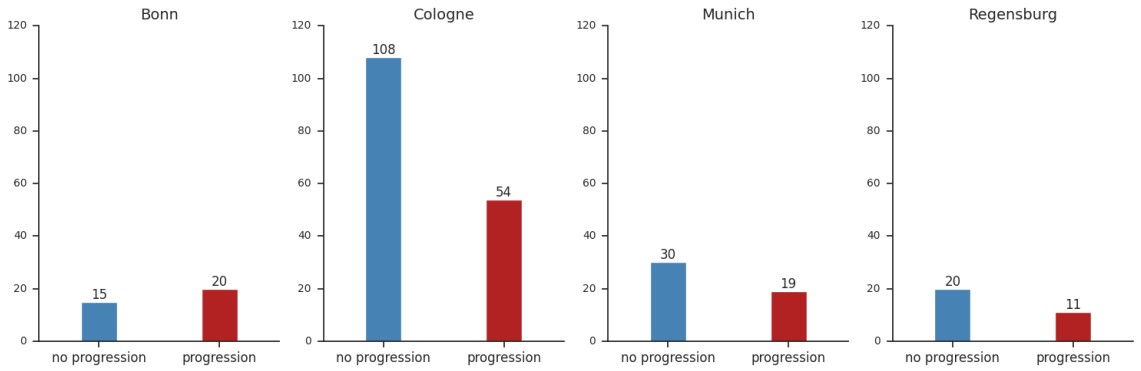

Figure 2: Labels' distribution in the cohorts from our dataset.

**Data preprocessing** Each WSI was tiled into patches of 256x256 pixels at 20x magnification. Patches without tissue were discarded, and remaining patches were embedded into their vector representations with an EfficientNet-v2-L model pre-trained on ImageNet, using the model's penultimate layer output before its classification head. Each WSI is then represented as a sequence that comprises all its feature vectors. For patients with multiple slides, we concatenated the sequences corresponding to their WSIs together, in order to facilitate making patient-level decisions.

We also tested including a stain normalization step with Reinhard's algorithm, (Reinhard et al., 2001) and replacing the EfficientNet-v2-L model with CTransPath (Wang et al., 2022), a transformer model trained without supervision on large histopathology dataset. However, none of these modifications improved the model's accuracy over the original pre-processing pipeline.

**Data heterogeneity** Figure 3 illustrates the visual diversity of the complete dataset. The projection is done using t-SNE computed on the average patch embedding of each patient (van der Maaten and Hinton, 2008). Data from each clinical center form a separate cluster with only limited overlap between the clusters, and each cluster is contained to a local region within the embedding space. This inter-center variability is a common phenomenon in quantitative WSI analysis, due to the differences in tissue processing steps, such as

staining duration, concentration or sample storage conditions (Bejnordi et al., 2014). The visual heterogeneity among the WSIs suggests that a single cohort dataset would not suffice to train a robust progression risk predictor, and highlights the importance of including data from diverse medical centers during training for improved model generalizability. We reached comparable findings by visualizing the data that had been processed using the alternative CTransPath feature extractor, and Reinhard's stain normalization (Figures 4 and 5.

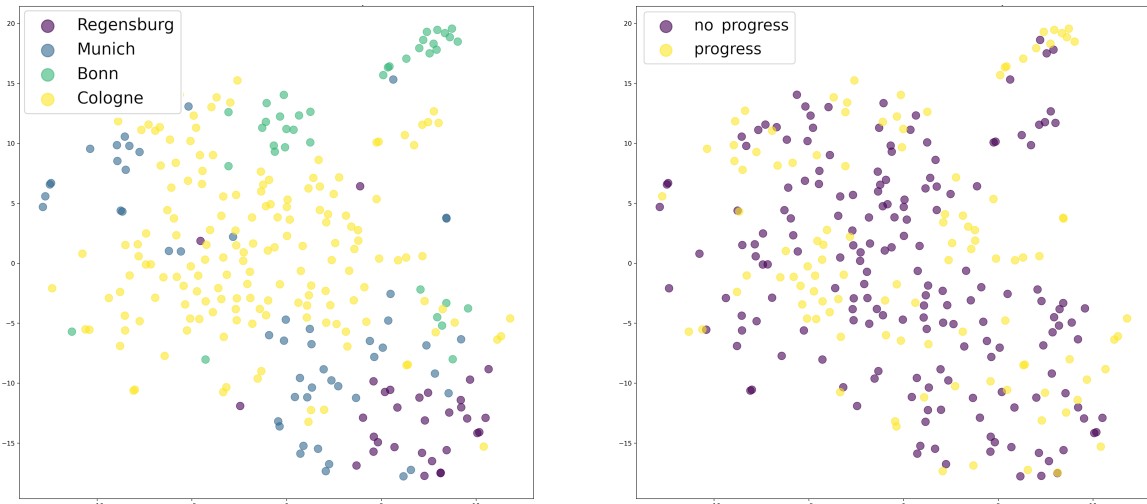

Figure 3: Visualization of data distribution of vectors obtained by preprocessing WSIs using EfficientNetV2 and reducing dimensionality with t-SNE. Data is colored per cohort (on the left) and per label (on the right). Data from each cohort cluster together, however, classes do not cluster.

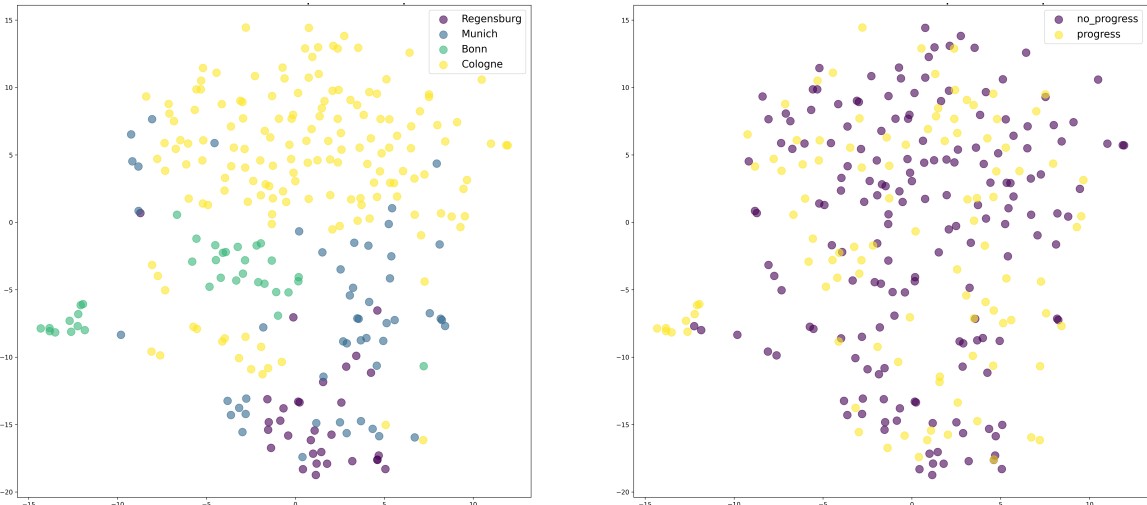

Figure 4: Visualization of data distribution of vectors obtained by preprocessing WSIs using CTransPath.

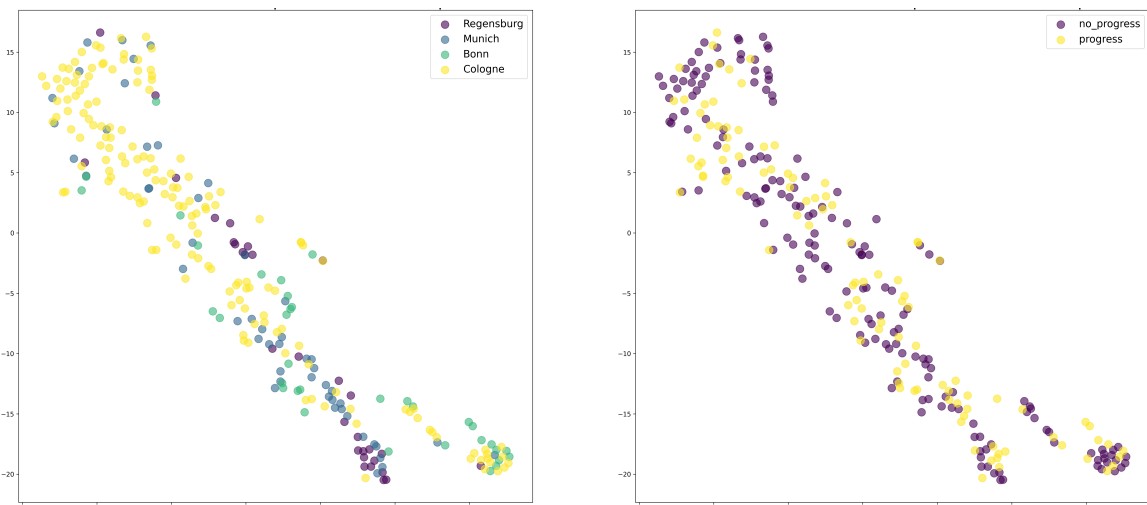

Figure 5: Visualization of data distribution of vectors obtained by preprocessing WSIs using EfficientNetV2 including Reinhard's normalization.

## Appendix B. Additional results

We have conducted additional preliminary experiments following a 70/10/20 train/val/test data splitting. We trained models with three FL strategies: FedAvg (McMahan et al., 2023), SuPerFed (Hahn et al., 2022), and FedAvg with additional local fine-tuning after federated training (denoted FedAvg+FT in our experiments). Additionally, we tried FedAvg scenarios including Reinhard stain normalization (Reinhard et al., 2001), and replacing the EfficientNet preprocessing model with a CTransPath feature extractor (Wang et al., 2022). Results are shown in Table 2. Evaluation of the FL frameworks revealed a significant performance disparity across the different methods. The standard federated averaging algorithm, FedAvg, exhibited poor performance, achieving the lowest accuracy, precision, recall, and F1-score among all evaluated methods. This highlights the challenges of effectively aggregating model updates in a federated setting without further refinement when data is non-independent and identically distributed (non-IID).

| Method | Accuracy | Precision | Recall | F1-score |
|---|---|---|---|---|
| Local models | 0.7876 | 0.7292 | **0.8458** | 0.7777 |
| **(FL)** FedAvg | 0.7018 | 0.6000 | 0.5000 | 0.4712 |
| **(FL)** FedAvg (CTransPath) | 0.6742 | 0.5179 | 0.4667 | 0.4909 |
| **(FL)** FedAvg (Reinhard norm.) | 0.6314 | 0.5667 | 0.4375 | 0.4364 |
| **(PFL)** SuPerFed | 0.8201 | **0.9058** | 0.7083 | 0.7590 |
| **(PFL)** FedAvg+FT | **0.8558** | 0.8559 | 0.8333 | **0.8146** |
| **(PFL)** FedAvg+FT (CTransPath) | 0.7128 | 0.6424 | 0.7292 | 0.6688 |
| **(PFL)** FedAvg+FT (Reinhard norm.) | 0.7876 | 0.8125 | 0.7750 | 0.7879 |

Table 2: Averaged prediction results metric over all methods used over a fixed split. The best results are marked in bold, second best are underlined.

Table 3 shows the results of the local model training for each of the four cohorts. Training details can be found in Appendix C. Results demonstrate the feasibility of training local cSCC progression risk predictors, achieving accuracies ranging from 0.7000 to 0.8788, with an average accuracy of 0.7876 across cohorts. The Cologne model exhibited superior performance across all evaluated metrics, attaining an accuracy of 0.8788 and precision and recall values of 0.8333. Notably, this is the largest cohort, comprising 57% of the total dataset, and the one exhibiting the largest class imbalance.

| Cohort | Accuracy | Precision | Recall | F1-score |
|---|---|---|---|---|
| Bonn | 0.7143 | 0.7500 | 0.7500 | 0.7500 |
| Cologne | 0.8788 | 0.8333 | 0.8333 | 0.8333 |
| Munich | 0.7000 | 0.6667 | 0.8000 | 0.7273 |
| Regensburg | 0.8571 | 0.6667 | 1.0000 | 0.8000 |

Table 3: Performance metrics of models trained locally.

| Method | Test Cohort | Accuracy | Precision | Recall | F1-score |
|---|---|---|---|---|---|
| **Local models** | Bonn | 0.60 | 0.60 | 0.82 | 0.64 |
| | Cologne | 0.90 | 0.89 | 0.78 | 0.83 |
| | Munich | 0.65 | 0.58 | 0.50 | 0.46 |
| | Regensburg | 0.65 | 0.33 | 0.40 | 0.33 |
| **FedAvg** | Bonn | 0.55 | 0.56 | 0.25 | 0.29 |
| | Cologne | 0.90 | 0.83 | 0.90 | 0.85 |
| | Munich | 0.65 | 0.20 | 0.13 | 0.16 |
| | Regensburg | 0.30 | 0.23 | 0.60 | 0.32 |
| **FedAvg+FT** | Bonn | 0.74 | 0.77 | 0.88 | 0.82 |
| | Cologne | 0.95 | 0.94 | 0.90 | 0.92 |
| | Munich | 0.78 | 0.78 | 0.51 | 0.58 |
| | Regensburg | 0.80 | 0.46 | 0.51 | 0.45 |

Table 4: Detailed evaluation of Table 1 per each cohort.

The best performing model of each category from Table 2 was then evaluated following 5-fold cross-validation, and these results are shown in Table 1. Table 4 displays center-level metrics of the models shown in Table 1.

## Appendix C. Model training details

Depending on the method used we have compared various hyperparameters combinations. In all the experiments to address the class imbalance we have applied class weights to address the class imbalance:

| | Class Weight | |
|---|---|---|
| **Cohort** | No Progression | Progression |
| Bonn | 1.17 | 0.88 |
| Cologne | 0.75 | 1.5 |
| Munich | 0.82 | 1.29 |
| Regensburg | 0.78 | 1.41 |

Table 5: Class weights assigned to comprise class imbalance

In the subsections below we present hyperparameters sets that we found most effective.

### C.1. Local training

- Optimizer: SGD

- Learning rate: 1.0e-4

- Weight decay: 1.0e-5

- We've used early stopping based on validation loss, the training usually required 30 to 40 epochs.

### C.2. FedAvg

- Optimizer: SGD

- Learning rate: 5.0e-4

- Weight decay: 5.0e-5

- Epochs: 10

- Rounds: 10

- Weights combining strategy: Proportional to cluster size.

### C.3. FedAvg+FT

- Optimizer: SGD

- Learning rate: 1.0e-4

- Weight decay: 1.0e-5

- Epochs (FL): 16

- Rounds: 32

- Epochs (FT): We've used early stopping based on validations loss, the fine-tuning usually required 10 to 40 epochs.

- Weights combining strategy: Proportional to cluster size.

### C.4. SuPerFed

- Optimizer: SGD

- Learning rate: 1.0e-3

- Weight decay: 1.0e-4

- Epochs (FL): 15

- Rounds: 6

- Personalization threshold: 3

## Appendix D. Evaluation metrics definitions

- True Positives (TP) - The number of instances correctly predicted as progression.

- True Negatives (TN) - The number of instances correctly predicted as no progression.

- False Positives (FP) - The number of instances incorrectly classified as progression.

- False Negatives (FN) - The number of instances incorrectly classified as no progression.

**Accuracy**    Fraction of samples correctly predicted.

$$\text{Accuracy} = \frac{\text{TP + TN}}{\text{TP + TN + FP + FN}}$$

**Precision**    Fraction of samples correctly predicted as progression to all labeled as progression.

$$\text{Precision} = \frac{\text{TP}}{\text{TP + FP}}$$

**Recall**    Fraction of progression samples that were retrieved.

$$\text{Recall} = \frac{\text{TP}}{\text{TP + FN}}$$

**F1-score**    Harmonic mean of precision and recall.

$$\text{F1 Score} = 2 \cdot \frac{\text{Precision} \cdot \text{Recall}}{\text{Precision + Recall}}$$

