# OpenReview forum: "Predicting Cutaneous Squamous Cell Carcinoma Progression Risk from Whole Slide Images with Federated Learning"
_MIDL.io/2025/Short_Papers — MIDL 2025 - Short Papers_

### Official Review · Reviewer_QMNc · 2025-04-28

**Rating:** 3
**Confidence:** 4

**Summary:**

This paper proposed evaluted a federated learning framework for the prediction of cSCC progression based on WSIs. The main idea is to use the TransMIL as local classifier and the federated learning is performed with FedAvg. Experiments on multi-site dataset across Germany demonstrated the effectiveness of the proposed method.

**Strengths:**

This is a good implementation of federated learning for cSCC prediction based on WSIs. It is happy to see that a simple FedAvg framework can enable the cross-site learning based on heteregenous representations. The detailed supplementary material can also help to understand the details of the framework as well as the comparisons/ablation studies. The overall pipeline is also clearly presented and easy to follow.

**Weaknesses:**

My concern is the numble of samples involved in this study. Compared with the general natural images, it is widely known that the collection is more challenging due to the ethical issues and limited cases. However, I am quite wondering whether the experiments with <300 samples to evaluate the performance of federated learning can really support the claims from the authors. It is expected to see the results with larger datasets.

---

### Decision · Program_Chairs · 2025-05-01

Accept